# Morphological Suitability Analysis of Urban Greenspaces with Rivers: A Case Study of the Lixiahe Riverine Area

**Xiao-Jun Wang** *,†, **Xiao Wei** †  **and Xin Chen**

Department of Landscape Architecture, School of Architecture, Southeast University, Nanjing 210095, China
* Correspondence: xjworking@163.com
† These authors contributed equally to this work.

**Abstract:** Retaining river channels and constructing waterfront greenspaces are the primary tasks of urban waterfront development in China. However, the natural characteristics of the water network are not fully considered in some urban greenspaces system planning and subsequent construction. We proposed a simple evaluation system to assess the morphological suitability between greenspaces and rivers in both the existing and planning stages. The evaluation indicators consist of two-level factors, in which the types of greenspace defined by the distance to the nearest river are the primary factors, including urban greenspace, waterfront greenspace and near-water greenspace, and the spatial forms of each type of greenspace are the secondary factors. The evaluation system can reflect the characteristics of each city and provide an overall comparison to cities of the same scale in similar regions. This study also investigated the impact of greenspace system planning on the current greenspace form. The results showed that near-water greenspace is a key factor that affects the matching degree among all primary factors, and the layout of greenspaces has a substantial impact on morphological suitability. Significant correlations between matching degree and evaluation factors were also found. This paper provides an in-depth understanding of urban greenspace form with urban rivers.

**Keywords:** urban greenspaces; urban greenspaces system planning; morphological suitability; matching degree; urban rivers; Lixiahe riverine area

## 1. Introduction

Most cities in the world are built near water networks that have rich natural resources. They not only provide essential life support for human beings but also shape the urban form and lay the foundation for the producing and living style [1]. The blue–green infrastructure of the urban water system and waterfront greenspaces has a profound impact on the urban artificial system [2] and contributes to the support of life, ecology, recreational activities and landscape needs [3,4].

As a significant part of urban ecological systems [5], water bodies contribute multiple ecological functions, including enhancing biodiversity [6], regulating urban climate [7–9] and regulating flooding [10]. Waterfront areas are usually regarded as the boundaries between land and aquatic systems, with unique landscape characteristics and "hotspots" of biological and chemical activities [11]. Research has shown a statistically significant relationship between amount and configuration of impervious areas and woodlands on a waterfront and a river's biological integrity in the lowland sub-basin [12]. Water bodies could enrich the bird community [13], and the green coverage of a riverbank is related to bird diversity along the riverside corridor [14]. In terms of regulating urban climate, studies have shown that a waterfront greenspace buffer zone has a significant cooling effect [15], and different structures and forms of waterfront greenspace in built-up areas have different synergistic cooling properties [7]. Specifically, blue–green space provides a higher cooling effect in summer, and the mean cooling extent and intensity are 150 m and 2.47 °C [16].

Meanwhile, river regulation and waterfront greenspace can improve detrimental conditions in severely degraded rivers, help to restore damaged river systems [17] and prevent eutrophication of adjacent waters [11].

Additionally, the water system is an important landscape and cultural resource. Regarding people's natural affinity to water edges, riverside greenspace can improve the quality of the living environment, create public space for residents and provide rich recreational and landscape value [18]. The linear riverbank in a waterfront greenspace is usually integrated with the urban greenway and other green infrastructure, becoming an entertainment and leisure area [19,20]. Waterfront space developed for or occupied by urban greenspace can shape the interface pattern of water/greenspace/buildings. This configuration often produces a better landscape effect because it is more layered and attractive. Moreover, rivers play an indispensable role in extended history. The Grand Canal in China was designated a World Cultural Heritage Site in 2014, which reflects ancient great wisdom integrated with social economy, culture, technology and landscape for thousands of years [21].

With rapid urbanization in China, however, urban development and construction have led to significant changes in the form of urban water systems and waterfront greenspaces, including shortening of river lengths, gradual disappearance of natural wetlands and water shoals [22], channelization of natural rivers, encroachment of greenspaces and a substantial reduction in riverbank vegetation [14,23]. With this trend, the negative impact of urban water environment deterioration has prompted development of substantial restoration practices. Urban river reconstruction and landscape projects have proliferated in the last 20 years, including water environment improvements, waterfront park construction and riverbank landscaping [3]. With the proposal of national territorial planning in China, blue–green space planning has become one of the key issues in the implementation of the ecological priority strategy [24], in which protecting the urban water system and developing waterfront greenspaces are common improvement strategies in urban river restoration projects. However, most of these waterfront greenspace construction projects are limited to the local- or site scale and scattered, often random and incomplete [25].

Therefore, the question becomes: for cities with a rich water network, what kind of urban greenspace construction is most conducive to the function of the urban water system? In other words, do we need to build riverside greenspaces with urban water network characteristics to form blue–green infrastructure with a higher matching degree? Is it necessary to have a long-term perspective on the ecological restoration of urban rivers and riverside greenspace construction during urban development? Therefore, at a city scale, analyzing the rationality of urban waterfront greenspace at both the existing and planning stages has become a priority when planning and designing urban greenspaces and construction activities.

Most current studies on urban waterfront greenspace emphasize the ecological benefits and restoration methods at the local- or site scale. Research on water systems at the city scale has been focused on evolution and characteristics of water system structures, connectivity, comprehensive planning management and restoration [26–28]. Furthermore, studies on the relationship between greenspaces and rivers mostly focus on their combined cooling effect [29–31], ecological function [32] and recreation services [33], in which the characteristics of rivers, spatial distribution and pattern of waterfront greenspaces are the main independent factors [7,29,34], while the interaction or suitability of greenspaces and rivers are underestimated. Moreover, few studies pay attention to the relationship from the perspective of planning and the overall features of the urban environment. This paper attempts to establish a preliminary analysis and evaluation system for greenspace form. In this case study, seven county-level cities with similar scale and river attributes in the Lixiahe area were selected, which is an important part of the Grand Canal adjacent area. The urban greenspaces and water systems were analyzed at both the existing and planning stages to examine the morphological suitability, or matching degree, between the urban greenspaces and water systems.

## 2. Materials and Methods

### 2.1. Research Framework

The research framework of this study is shown in Figure 1. The first step was data preprocessing: to identify the existing situation and the future planning of greenspaces and water systems in the study area. This was based on GF-1 remote sensing images and urban planning maps issued by the government. Second, a two-level indicator system was proposed to evaluate the matching degree between greenspaces and water systems. A comparison of the matching degree between seven cities and between the existing situation and future planning was conducted. Finally, a discussion of possible reasons for the differences and optimized strategies that could guide urban greenspace system planning for cities with dense water networks was undertaken.

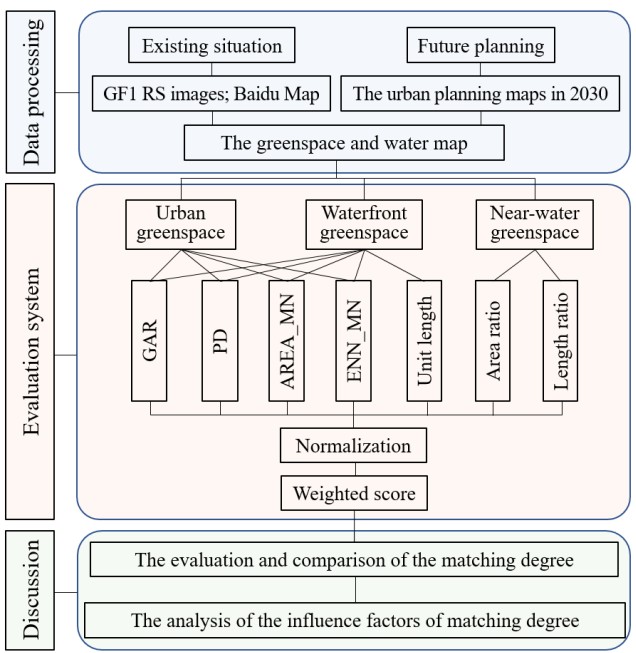

**Figure 1.** Research framework. (Note: GAR is short for greenspace area ratio; PD is short for patch density; AREA_MN is short for mean patch area; ENN_MN is short for mean Euclidean nearest neighbor distance).

### 2.2. Study Area

Seven county-level cities with similar size and river attributes as well as the same urbanization level in the Lixiahe area were selected as case cities. The Lixiahe area is located in the middle of Jiangsu Province, China. It is a dished plain depression centered in Xinghua City with highlands and several small-dished depressions. It was originally an ancient lagoon, formed by accumulation of sediment from the Yangtze River, Huaihe River and Yellow River. The overall terrain is high in the east and low in the west. This area is a wetland of the Huaihe River Basin, starting from the Liyun Canal in the west, to the Chuanchang River in the east and from the general irrigation canal in Northern Jiangsu in the north and the New Tongyang canal in the south. It is about 32°25′–34°07′ N and 119°08′–120°56′ E. It has a total area of more than 13,500 square kilometers and a population of more than 10 million. The Lixiahe area is part of a transition zone from a subtropical to a warm temperate zone, a climate conducive to monsoons, sufficient sunshine and four distinct seasons. The annual average temperature is 14–15 °C, and the frost-free period is 210–220 days. The average annual precipitation is 1000 mm, and the average annual evaporation is approximately 960 mm. Seven representative small- and medium-sized cities with abundant water systems in the Lixiahe area were selected for the comparison. The study area of each city is the main built-up area, in which the canals and rivers are both

included in the water system as they are all well-developed for years as a part of urban infrastructure. Meanwhile, the widths of canals and rivers through those cities are in the same order that is appropriate to compare. General information for each city is shown in Table 1 below. The locationss of the Lixiahe area and seven selected cities are shown in Figure 2.

**Table 1.** General information of the selected cities.

|  | Xinghua | Gaoyou | Dongtai | Jianhu | Baoying | Jiangyan | Haian |
|---|---|---|---|---|---|---|---|
| Population (unit: 10,000) | 155.67 | 80.26 | 109.81 | 80.06 | 87.97 | 74.35 | 86.30 |
| Study area (unit: km$^2$) | 46.82 | 33.89 | 51.41 | 42.01 | 48.34 | 45.29 | 44.51 |
| Total area of urban greenspace (unit: km$^2$) | 9.52 | 3.35 | 8.82 | 12.52 | 15.35 | 6.73 | 9.28 |
| Total area of water system (unit: km$^2$) | 5.57 | 4.63 | 4.77 | 2.80 | 3.96 | 1.90 | 2.86 |
| Total length of water system (km) | 221.86 | 190.62 | 278.91 | 183.56 | 298.85 | 142.86 | 164.41 |

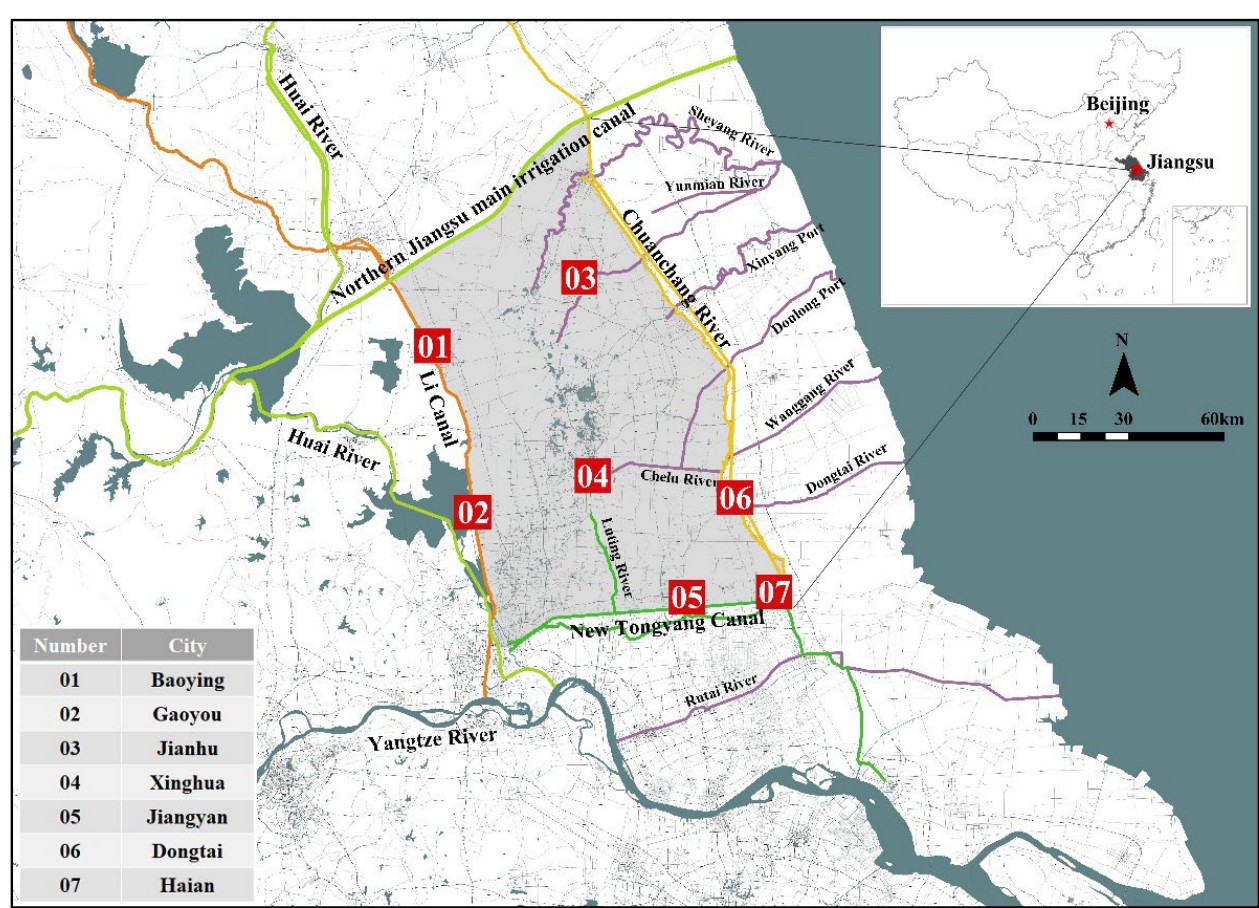

| Number | City |
|---|---|
| 01 | Baoying |
| 02 | Gaoyou |
| 03 | Jianhu |
| 04 | Xinghua |
| 05 | Jiangyan |
| 06 | Dongtai |
| 07 | Haian |

**Figure 2.** Location of the study area.

### 2.3. Data Collection and Processing

This study evaluated the matching degree of greenspaces and water systems at both the existing and planning stages of each city. The existing data were extracted according to GF-1 remote sensing images and calibrated by the Baidu Map (https://map.baidu.com accessed on 20 October 2020). The planning data were drawn based on the 2030 Urban Master Plan published by the municipal government of each city.

The existing data contain four scenes GF-1 remote sensing images in August and September 2020, with a panchromatic image resolution of 2 m and a multispectral image resolution of 8 m. The details of each image and cities included are shown in Table 2. ENVI V5.3.1 software (Exelis Visual Information Solutions, Broomfield, CO, USA) was

used to preprocess the remote sensing data, including atmospheric correction, radiometric calibration, image fusion and spatial subset. The main built-up areas of each city were selected as the primary research area. In this paper, greenspaces and waterbodies were extracted using NDVI and NDWI indexes, and then repeatedly corrected with the Baidu Map (https://map.baidu.com/ accessed on 20 October 2020) to determine the final extraction thresholds for each city. Calculations of the two indexes and selection thresholds for each city are shown in Table 3.

**Table 2.** Information of the selected GF-1 images.

| Numbers | Product Type | Acquisition Time | Cloud Cover | Covered Cities |
|---|---|---|---|---|
| 1 | GF1-B/PMS | 5 September 2022 11:29:06 | 5% | Gaoyou, Baoying, Xinghua |
| 2 | GF1-B/PMS | 3 August 2020 11:24:54 | 1% | Jianhu |
| 3 | GF1-D/PMS | 18 August 2020 11:05:19 | 1% | Dongtai |
| 4 | GF1-D/PMS | 18 August 2020 11:05:28 | 1% | Haian, Jiangyan |

**Table 3.** Calculation of the indexes and thresholds for each city.

| Calculation Method | Thresholds | | | | | | |
|---|---|---|---|---|---|---|---|
| | Xinghua | Gaoyou | Dongtai | Jianhu | Baoying | Jiangyan | Haian |
| $NDVI = (NIR - RED)/(NIR + RED)$ | 0.45 | 0.42 | 0.50 | 0.60 | 0.50 | 0.45 | 0.45 |
| $NDWI = (GREEN - NIR)/(GREEN + NIR)$ [1] | >−0.05 | >−0.15 | >−0.23 | >−0.23 | >−0.15 | >−0.15 | >−0.25 |

[1] GREEN, RED and NIR correspond to the value of Band 2, 3 and 4 of GF-1 image, respectively.

The planning data were based on the land use classification published in the 2030 Urban Master Plan by each government's department. With the help of the reclassification tool in GIS, the greenspaces and waterbodies were extracted as well.

*2.4. The Evaluation System of Matching Degree between Greenspaces and Water Systems*

2.4.1. Selection of Evaluation Indicators

The evaluation indicators to evaluate the morphological matching degree of greenspaces and water systems were divided into two levels. The primary indicator was the type of greenspace, which was further divided into three components: urban greenspace (UGS), waterfront greenspace (WGS) and near-water greenspace (NGS). The UGS reflects the overall condition of greenspaces in the city. The WGS along the river is an important aspect of urban waterfront space. It reflects the combination of the water network in the greenspace and the natural ecological environment along the river. In comparison, the NGS is more important. This part of greenspace is directly adjacent to the water bodies, which is usually regarded as the expansion of the natural space of water bodies. The definitions and illustrations of the three types of greenspace are as shown in Table 4 below.

The secondary indicators consisted of three components. The first was the greenspace area ratio (GAR), which is the proportion of greenspace in the study area. This is the most commonly used indicator in urban greenspace planning and evaluation in China [35]. The second component was comprised of the density, size and distribution of greenspace patches, quantified by three landscape metrics: patch density (PD), mean patch area (AREA_MN) and mean Euclidean nearest neighbor distance (ENN_MN). These metrics are commonly used in landscape ecology and have significant relationship with landscape design [36,37]. Third, focusing on the water system, the matching degree between greenspaces and water systems was analyzed. This includes the unit length of WGS patches per length of water system (Unit_leng), the area ratio (AR) and length ratio (LR) of NGS in WGS. Because of the different emphasis of each type of greenspace, the evaluation of UGS included the first 4 indicators, the evaluation of WGS included the first 5 indicators and the evaluation of NGS included the last 2 indicators. The calculation method of each secondary indicator is shown in Table 5.

**Table 4.** Definitions and Illustrations of the three types of greenspace.

| Types | Definitions | Illustrations |
|---|---|---|
| Urban greenspace | All types of greenspace in the city. |  |
| Waterfront greenspace | The greenspace located within the 100 m buffer zone of water system, and the part of the patch beyond the buffer zone was deleted. |  |
| Near-water greenspace | The greenspace located within the 100 m buffer zone of water system and directly adjacent to the water bodies, and the turning or extension part not along the water bodies was deleted. |  |

**Table 5.** Calculation of secondary indicators.

| Primary Indictors | Secondary Indicators | Calculation Method |
|---|---|---|
| UGS & WGS | GAR | The proportion of greenspaces in the study area $\text{GAR} = A_g/A$, where $A_g$ is the total area of greenspaces and $A$ is the total area of the study area (unit: %) |
| | PD | The ratio of the number of greenspace patches of the study area, $\text{PD} = n/A$ where n is the number of patches, $A$ is the total area of the built-up area (unit: number per km$^2$) [38] |
| | AREA_MN | Average area of greenspace patches, $\text{AREA}_{MN} = \frac{\sum_1^n A_i}{n}$ where $n$ is the number of patches, $A_i$ is the area of the $i$-th patch (unit: ha) [38] |
| | ENN_MN | Mean distance to the nearest neighboring patch of the same type based on the edge-to-edge distance, $\text{ENN\_MN} = \sum_1^n h_i$ where n is the number of patches, $h_i$ is the distance of the i-th patch to the nearest patch (unit: m) [38] |

**Table 5.** *Cont.*

| Primary Indictors | Secondary Indicators | Calculation Method |
|---|---|---|
| WGS | Unit_leng | The length of WGS patches along the river near the water system per unit length, Unit_leng $= P_g / P_w$, where $P_g$ is the total length of WGS. Here, only the length of green patches along the river was calculated, and $P_w$ is the total length of the water system (unit: m) |
| NGS | AR | The area ratio of NGS in WGS, AR $= A_{NGS} / A_{WGS}$, where $A_{NGS}$ is the total area of NGS and $A_{WGS}$ is the total area of WGS (unit: %) |
| | LR | The length ratio of NGS in WGS, LR $= P_{NGS} / P_{WGS}$, where $P_{NGS}$ is the total length of NGS and $P_{WGS}$ is the total length of WGS (unit: %) |

### 2.4.2. The Normalization Method

Since different indicators have different units and orders of magnitude, all indicators were normalized for subsequent comparisons. The normalization method was as follows:

$$N_{Index_{ij}} = Index_{ij} / \frac{\sum_1^n Index_i}{n} \tag{1}$$

where $Index_{ij}$ is the *j*-th index of the *i*-th city; *n* is the number of cities (*n* = 7).

Among all indicators, the lower the actual value of ENN_MN, the better effect of the greenspace distribution. Therefore, to ensure that all indicator scores were the same sequence with its actual evaluation results, the reciprocal of the normalized score for ENN_MN was taken as its evaluation score. Then, the sum of the corresponding indicators was the score of the matching degree for that type of greenspace. Further, Spearman correlation coefficients were calculated to investigate the relationship among all the indicators.

### 2.4.3. The Weighted Score

Due to the varying importance of each type of greenspace, different weights were obtained according to the interaction degree between the greenspace and water system, which was 20% for UGS, 30% for WGS and 50% for NGS. The matching degree scores for the various types of greenspace were weighted and then added together to obtain the matching degree score for each city.

## 3. Results

### 3.1. Evaluation of Matching Degree of Greenspaces in the Existing Stage

The matching degree scores between greenspaces and water systems at the existing stage are shown in Figure 3. Among the UGS indicators, there were slight differences in mean Euclidean nearest neighbor distance (ENN_MN) between the seven cities. The differences were mainly reflected in greenspace area ratio (GAR), patch density (PD) and mean patch area (AREA_MN). The rankings for GAR and AREA_MN were basically the same, among which Jianhu and Baoying were the highest; Gaoyou and Jiangyan were the lowest. The ranking for ENN_MN was similar to those of GAR and AREA_MN, but the range was small, among which Xinghua was the highest and Gaoyou the lowest. The PD scores for UGS were the opposite of the other three indicators. Gaoyou and Xinghua had the highest scores, while Jianhu and Baoying had the lowest.

As for WGS, the rankings for PD and AREA_MN were completely opposite, and there were large differences between cities. Specifically, Gaoyou had the highest PD and the smallest AREA_MN, while Baoying had the lowest PD and the largest AREA_MN. With regard to GAR, Gaoyou scored the highest, followed by Jianhu and Dongtai, which scored the lowest. There were slight differences among the other cities. For the ENN_MN and unit length (Unit_leng) indicators, except for Xinghua, which had both highest scores on

these two indicators, the score rankings of those two indicators for the other six cities were just the opposite.

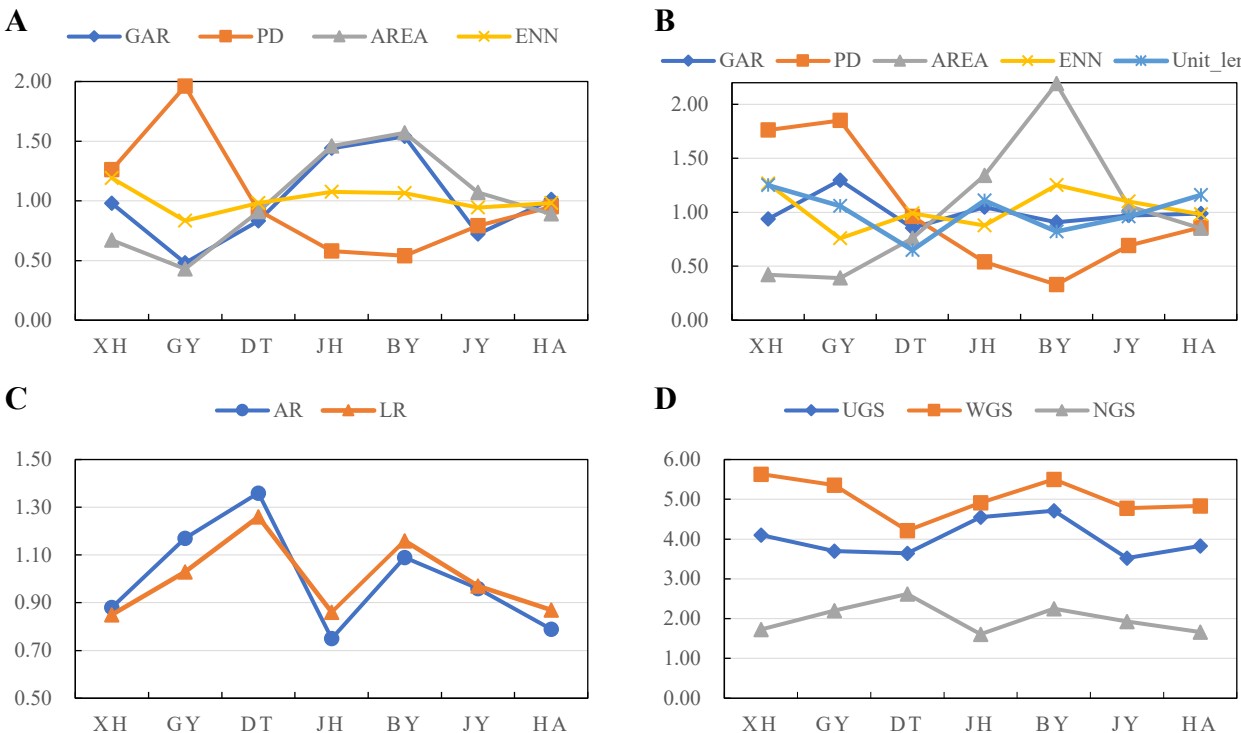

**Figure 3.** Matching degree scores of urban greenspace (**A**), waterfront greenspace (**B**) and near-water greenspace (**C**) and the total score of three types of greenspace (**D**) in the existing stage. (Note: XH is short for Xinghua, GY for Gaoyou, DT for Dongtai, JH for Jianhu, BY for Baoying, JY for Jiangyan, HA for Haian. GAR is short for greenspace area ratio; PD is short for patch density; AREA_MN is short for mean patch area; ENN_MN is short for mean Euclidean nearest neighbor distance; Unit_leng is short for unit length; AR is short for area ratio; LR is short for length ratio).

Among the indicators for NGS, two indicator scores for the seven cities were very close, among which Dongtai had the highest, followed by Baoying and Gaoyou, while Jianhu had the lowest score.

In a word, the score rankings for UGS and WGS were basically the same, while the ranking for NGS was basically the opposite of UGS and WGS. In the evaluation of UGS, Xinghua and Baoying scored the highest, Dongtai the lowest, while the scores for the other four cities were close. In the evaluation of WGS, Jianhu and Baoying scored the highest, followed by Xinghua and Jiangyan, which scored the lowest. In the evaluation of NGS, Dongtai scored the highest, followed by Gaoyou, Baoying and Jianhu, which scored the lowest.

### 3.2. Evaluation of Matching Degree of Greenspace in Future Planning

The matching degree scores between greenspaces and water systems in the planning stage are shown in Figure 4. As for UGS, the rankings for GAR and AREA_MN were basically the same, among which Baoying, Xinghua and Haian had better evaluation scores on these two indicators. The rankings for PD were opposite to those of GAR and AREA_MN. Gaoyou, Xinghua and Jianhu had higher scores. The scores for ENN_MN varied greatly among several cities, among which Jianhu had the highest score and Dongtai the lowest.

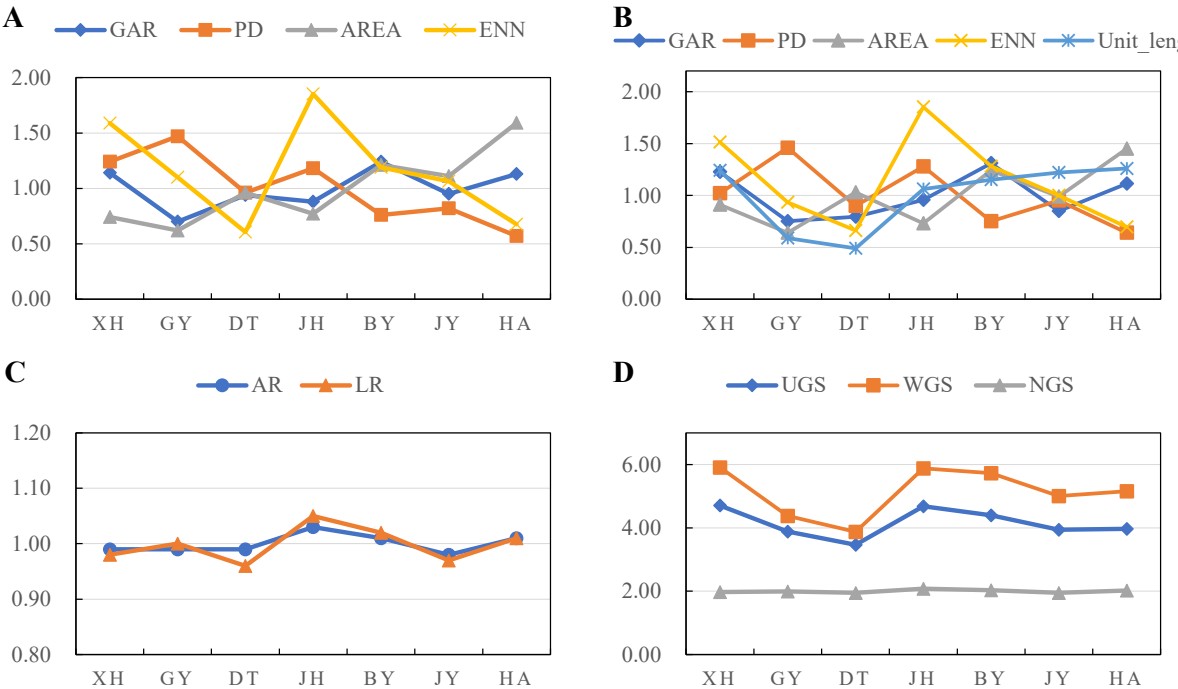

**Figure 4.** Matching degree scores of urban greenspace (**A**), waterfront greenspace (**B**) and near-water greenspace (**C**) and the total score of three types of greenspace (**D**) in the planning stage. (Note: XH is short for Xinghua, GY for Gaoyou, DT for Dongtai, JH for Jianhu, BY for Baoying, JY for Jiangyan, HA for Haian. GAR is short for greenspace area ratio; PD is short for patch density; AREA_MN is short for mean patch area; ENN_MN is short for mean Euclidean nearest neighbor distance; Unit_leng is short for unit length; AR is short for area ratio; LR is short for length ratio).

The rankings for PD and AREA_MN of WGS were just the opposite. The PD scores for Gaoyou and Jianhu were higher, while AREA_MN scores were lower. The PD scores for Baoying and Haian were the lowest, while AREA_MN scores were the highest. This trend was similar to the rankings in the existing stage, but the gap in evaluation scores was reduced. For GAR, Xinghua and Baoying scored higher, while Gaoyou and Jiangyan scored lower. The index score for ENN_MN green patches along the river varied greatly among cities, and the scores for Xinghua and Jianhu were the highest, while the scores for Dongtai and Haian were the lowest. As for Unit_leng, Gaoyou and Dongtai had the lowest scores, while the other five cities had relatively high scores.

The scores for NGS were basically the same in all seven cities, indicating that all cities considered NGS in future planning and the overall matching degree was similar.

As for the overall scores, Xinghua and Jianhu scored the highest and Dongtai the lowest in the evaluation of UGS and WGS. The total score rankings for WGS and UGS in the different cities were basically the same.

*3.3. Comparison of Morphological Matching Degree between Existing Stage and Future Planning*

3.3.1. Urban Greenspace

The comparison of UGS between the existing stage and future planning is shown in Table 6. Further, the change types of UGS with corresponding planning scheme were divided into the following three categories, as shown in Table 7.

**Table 6.** Comparison of matching degree between existing stage and future planning.

| The First Indicators | The Secondary Indicators | Xinghua | | Gaoyou | | Dongtai | | Jianhu | | Baoying | | Jiangyan | | Haian | |
|---|---|---|---|---|---|---|---|---|---|---|---|---|---|---|---|
| | | Future | Existing | Future | Existing | Future | Existing | Future | Existing | Future | Existing | Future | Existing | Future | Existing |
| UGS | GAR | 1.14 | 0.98 | 0.70 | 0.48 | 0.94 | 0.83 | 0.88 | 1.44 | 1.24 | 1.54 | 0.95 | 0.72 | 1.13 | 1.01 |
| | PD | 1.24 | 1.26 | 1.47 | 1.96 | 0.96 | 0.92 | 1.18 | 0.58 | 0.76 | 0.54 | 0.82 | 0.79 | 0.57 | 0.95 |
| | AREA_MN | 0.74 | 0.67 | 0.62 | 0.43 | 0.96 | 0.91 | 0.77 | 1.46 | 1.21 | 1.57 | 1.11 | 1.07 | 1.59 | 0.89 |
| | ENN_MN | 1.59 | 1.19 | 1.10 | 0.83 | 0.61 | 0.98 | 1.85 | 1.08 | 1.19 | 1.06 | 1.06 | 0.94 | 0.68 | 0.98 |
| WGS | GAR | 1.23 | 0.94 | 0.75 | 1.30 | 0.79 | 0.85 | 0.96 | 1.05 | 1.31 | 0.91 | 0.85 | 0.97 | 1.11 | 0.99 |
| | PD | 1.02 | 1.76 | 1.46 | 1.85 | 0.90 | 0.96 | 1.28 | 0.54 | 0.75 | 0.33 | 0.95 | 0.69 | 0.64 | 0.86 |
| | AREA_MN | 0.91 | 0.42 | 0.64 | 0.39 | 1.03 | 0.76 | 0.73 | 1.34 | 1.24 | 2.19 | 0.99 | 1.06 | 1.45 | 0.85 |
| | ENN_MN | 1.52 | 1.27 | 0.93 | 0.76 | 0.66 | 0.99 | 1.85 | 0.88 | 1.28 | 1.25 | 1.00 | 1.10 | 0.69 | 0.98 |
| | Unit_leng | 1.24 | 1.25 | 0.59 | 1.06 | 0.49 | 0.65 | 1.06 | 1.11 | 1.15 | 0.82 | 1.22 | 0.96 | 1.26 | 1.16 |
| NGS | AR | 0.99 | 0.88 | 0.99 | 1.17 | 0.99 | 1.36 | 1.03 | 0.75 | 1.01 | 1.09 | 0.98 | 0.96 | 1.01 | 0.79 |
| | LR | 0.98 | 0.85 | 1.00 | 1.03 | 0.96 | 1.26 | 1.05 | 0.86 | 1.02 | 1.16 | 0.97 | 0.97 | 1.01 | 0.87 |

Note: GAR is short for greenspace area ratio; PD is short for Patch Density; AREA_MN is short for Mean Patch Area; ENN_MN is short for Mean Euclidean Nearest Neighbor Distance; Unit_leng is short for unit length; AR is short for area ratio; LR is short for length ratio.

**Table 7.** Types of UGS comparisons between existing stage and future planning.

| The Change in Indicators | | | | The Planning Scheme | Cities |
|---|---|---|---|---|---|
| GAR | PD | AREA_MN | ENN_MN | | |
| ↑ [1] | ↑ | ↑ | ↑ | Increased the number and area of green patches and compact distribution. | Jiangyan |
| | | | ↓ | Increased the number and area of green patches and scattered distribution. | Dongtai |
| ↑ | ↓ | ↑ | ↑ | Reduced the number of green patches; increased the area of patches; integrated multiple small patches into larger patches or canceled the original small green patches and planned large green patches and compact distribution. | Xinghua, Gaoyou |
| | | | ↓ | Reduced the number of green patches; increased the area of patches; integrated multiple small patches into larger patches or canceled the original small green patches; planned large green patches and scattered distribution. | Haian |
| ↓ | ↑ | ↓ | ↑ | Increased the number of green patches; reduced the area of patches and divided the original green patches into smaller green patches with compact distribution. | Jianhu, Baoying |

Note: [1] the symbols ↑ and ↓ indicate the rise and fall, respectively, of planning stage scores compared with the existing stage. GAR is short for greenspace area ratio; PD is short for patch density; AREA_MN is short for mean patch area; ENN_MN is short for mean Euclidean nearest neighbor distance.

The first three indicators for UGS in Dongtai improved to some extent, but the increases in PD and AREA_MN were small, while ENN_MN increased substantially. That is, the greenspace in the future planning was more scattered. The GAR for Jiangyan in future planning was higher than in the existing stage, but the growth rates of PD, AREA_MN and ENN_MN for Jiangyan were relatively small, mainly due to the increasing number and area of patches and improvement in the greenspace ratio.

Compared with the existing stage, the future planning for Xinghua and Gaoyou had larger AREA_MN, slightly lower PD and higher ENN_MN scores. That is, a small part of the greenspace had been removed in the future planning, but the area of retained greenspace patches increased. The average distance between greenspaces became closer, and the overall greenspace rate improved. The AREA_MN for Haian increased substantially, and the scores for PD and ENN_MN decreased a great deal, indicating that the future plan for Haian was to merge the existing small patches to form large green patches, or cancel some small green patches and focus on the construction of large green patches. Although the overall number of green patches became smaller, the area of each patch increased, so the total amount of greenspace increased.

Although Jianhu and Baoying had improved scores for PD and ENN_MN, AREA_MN decreased. In other words, although the number of green patches increased and the average distance between patches had been shortened, the average area of green patches actually became smaller, showing a fragmented trend, resulting in a reduction in the total amount of greenspace.

### 3.3.2. Waterfront Greenspace

The comparison of the future planning and existing situations of WGS in each city was divided into the following scenarios (see Table 8).

**Table 8.** Types of WGS comparisons between existing stage and future plans.

| Scenarios | GAR | PD | AREA_MN | ENN_MN | Unit_leng | City |
|:---:|:---:|:---:|:---:|:---:|:---:|:---:|
| | ↓[1] | ↓ | ↑ | ↓ | ↓ | Dongtai |
| 1 | ↑ | | | | ↑ | Haian |
| | ↓ | | | | ↓ | Jianhu |
| | ↑ | ↑ | ↓ | ↑ | ↑ | Baoying |
| | ↓ | ↓ | ↑ | ↓ | ↓ | Dongtai |
| 2 | ↑ | | | ↑ | | Xinghua |
| | ↓ | ↑ | ↓ | ↓ | ↑ | Jiangyan |
| | ↑ | | | ↑ | ↑ | Baoying |
| | ↑ | ↓ | ↑ | ↓ | ↑ | Haian |
| 3 | ↑ | | | ↑ | ↓ | Xinghua |
| | ↓ | ↑ | ↓ | ↑ | ↓ | Jianhu |
| | | | | ↓ | ↑ | Jiangyan |
| 4 | ↓ | ↓ | ↑ | ↑ | ↓ | Gaoyou |
| | | ↑ | ↓ | | | Jianhu |
| | ↓ | ↓ | ↑ | ↑ | ↓ | Gaoyou |
| 5 | ↑ | | | | ↓ | Xinghua |
| | ↓ | | | ↑ | | Gaoyou |
| | | | ↑ | | ↓ | Dongtai |

Note: [1] the symbols ↑ and ↓ indicate the rise and fall, respectively, of planning scores compared with the existing stage. GAR is short for greenspace area ratio; PD is short for patch density; AREA_MN is short for mean patch area; ENN_MN is short for mean Euclidean nearest neighbor distance; Unit_leng is short for unit length.

When changes in the PD, AREA_MN and ENN_MN scores were the same, then changes in the GAR and Unit_leng scores showed a significant positive correlation ($r = 0.562$, $p < 0.05$). The representative cities were Dongtai and Haian and Jianhu and Baoying. Based on the calculations of these two indicators, it can be inferred that the relative changes in the total area and total length of WGS were consistent when the future planning and existing situations of the four cities were compared. The total study area and length of the water systems were basically unchanged. That is, these two indicators increased or decreased simultaneously.

When changes in the PD, AREA_MN and Unit_leng scores were the same, then GAR scores changed consistently with the changes in ENN_MN scores. The representative cities were Dongtai and Xinghua and Jiangyan and Baoying. This shows that, compared with the planning and existing stages of the four cities, the GAR for WGS was consistent with the relative changes in ENN_MN. That is, the closer the distance between patches, the higher the possible GAR.

When changes in the GAR, PD and AREA_MN scores were the same, then changes in ENN_MN scores are opposite to Unit_leng scores. The representative cities were Haian and Xinghua and Jianhu and Jiangyan. This revealed that, when comparing the future planning and existing stage of the four cities, the closer the greenspace patches were to each other, the Unit_leng may be increased simultaneously.

When changes in the GAR, ENN_MN and Unit_leng were the same, then changes in PD and AREA_MN showed a significant negative correlation (r = −0.956, *p* < 0.01). The representative cities were Gaoyou and Jianhu. Comparing these two cities, only the changes in PD and AREA_MN indicators were different, and the changes in the indicators in the same cities were opposite, revealing that the differences of those two cities are mainly reflected in the density and size of greenspace patches.

In addition, only the change in the score of GAR was different between Gaoyou and Xinghua, and only the change in ENN was different between Gaoyou and Dongtai. According to the second type mentioned above, it can be concluded that the GAR for Dongtai and Xinghua may have similar change with ENN_MN. In reverse, it can be inferred that the GAR for Gaoyou has opposite change with ENN_MN. That is, when the GAR increases, ENN_MN may decrease.

### 3.3.3. Near-Water Greenspace

The area ratio (AR) and length ratio (LR) of NGS in Xinghua, Jianhu and Haian along the river increased. The AR in Jiangyan increased slightly, and LR remained unchanged. The AR and LR of NGS in Gaoyou, Dongtai and Baoying along the river decreased. Xinghua, Jianhu and Haian increased the ratio of NGS in the plans for WGS to improve the near-water experience of users by adding more greenspace along the river, while Jiangyan, Gaoyou, Dongtai and Baoying developed greenspaces more perpendicular to the river.

## 4. Discussion

In our research on planning practices of urban greenspaces in this region, we found that urban greenspace system planning and waterfront greenspace development in some cities did not fully consider and follow the natural characteristics of the area. Most of the urban greenspace construction was carried out in local river sections and often lacked integrity at the city level, which occasionally led to the embarrassing situation that the greenspace conditions in some cities became worse in the planning stage than in the existing stage. Here, two typical examples (EP-1 & EP-2) are selected to illustrate different types of waterfront spaces in Table 9, in which EP-1 is well developed from the perspective of matching degree between water and greenspace, while EP-2 is not so good. Seven different types of waterfront spaces are categorized with descriptions and colored in the corresponding area of the example to further explain the issue.

**Table 9.** Examples and Illustrations of different waterfront space types.

| Examples | |
|---|---|
| 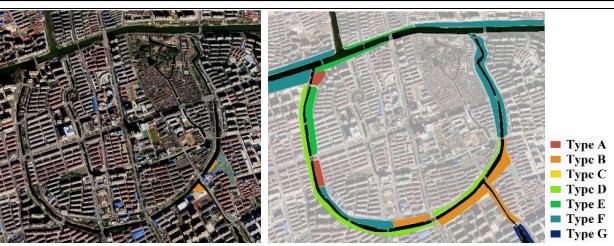 | 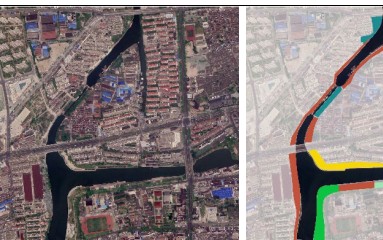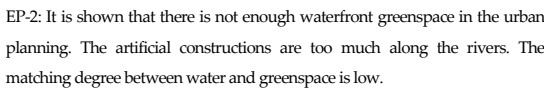 |
| EP-1: It is shown that the waterfront greenspaces are well developed along the main rivers. Most of the waterfront spaces are retained as greenery and recreation. The road planning leaves enough space to the greenspaces. The types of waterfront greenspace are diverse. | EP-2: It is shown that there is not enough waterfront greenspace in the urban planning. The artificial constructions are too much along the rivers. The matching degree between water and greenspace is low. |

| Types | 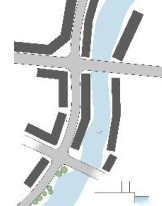 | 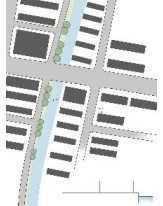 | 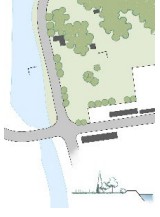 | 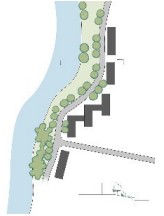 | 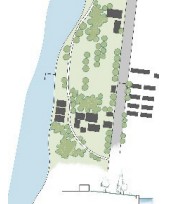 | 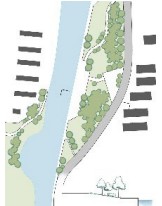 | 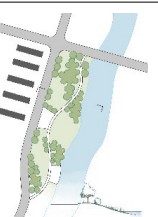 |
|---|---|---|---|---|---|---|---|
| Descriptions | A. Dense buildings are perpendicular to the river, and there is only a little vegetation as revetment along the roads. | B. Dense buildings along the river, and there is little waterfront greenspace. | C. Main roads are along the river with narrow greenspaces between roads and river, and the greenspaces on the other side of roads have no interaction with river because of the barrier of roads. | D. There are linear greenspaces less than 12 m between the main roads and river. The revetments are made by hard materials. | E. There are wide and open greenspaces along the river, with abundant vegetation, architecture and recreational space. The revetments are made by hard materials. | F. There are wide and open greenspaces along the river, with abundant vegetation and no architecture. The revetments are made by hard materials. | G. There are wide and open greenspaces along the river, with abundant vegetation and water-enjoyable space. The revetments are made by ecological materials, which can improve the ecological quality of waterfront space. |

*4.1. Main Findings and Significance of the Study*

Based on the framework established in this study and the scores for the existing stages and future plans, we were able to analyze the change in greenspace form of seven cities. There were three primary indicators that allowed examination of whether the impact of future plans on current greenspace was positive or negative. We then identified the factors that caused these changes based on further analysis of the secondary indicators. At the same time, we also compared the overall change in greenspace between all the cities and discussed which factors caused the differences between these cities. The main findings and discussion of the paper are as follows:

4.1.1. Overall Evaluation of Matching Degree

In the evaluation process, we used the concept of matching degree (MD) to evaluate the morphological suitability of greenspace form. MD is a weighted average sum of rating scores of the three greenspace categories, which included the existing stage (MD e), planning stage (MD p) or in both situations (MD e+p). See Table 10.

**Table 10.** Scores and rankings of existing and planning greenspaces.

| Stages | Primary Indicators | Xinghua | Gaoyou | Dongtai | Jianhu | Baoying | Jiangyan | Haian |
|---|---|---|---|---|---|---|---|---|
| | UGS | 4.10 | 3.70 | 3.64 | 4.56 | 4.71 | 3.52 | 3.83 |
| | WGS | 5.63 | 5.35 | 4.21 | 4.91 | 5.50 | 4.78 | 4.84 |
| Existing | NGS | 1.73 | 2.2 | 2.62 | 1.61 | 2.25 | 1.93 | 1.66 |
| | MD e | 3.37 | 3.45 | 3.30 | 3.19 | 3.72 | 3.10 | 3.05 |
| | ranking | 3 | 2 | 4 | 5 | 1 | 6 | 7 |
| | UGS | ↑[1] 4.71 | ↑ 3.89 | ↓ 3.47 | ↑ 4.68 | ↓ 4.40 | ↑ 3.94 | ↑ 3.97 |
| | WGS | ↑ 5.91 | ↓ 4.38 | ↓ 3.88 | ↑ 5.88 | ↑ 5.73 | ↑ 5.01 | ↑ 5.16 |
| Planning | NGS | ↑ 1.97 | ↓ 1.99 | ↓ 1.95 | ↑ 2.08 | ↓ 2.03 | ↑ 1.95 | ↑ 2.02 |
| | MD p | 3.70 | 3.09 | 2.83 | 3.74 | 3.61 | 3.27 | 3.35 |
| | ranking | 2 | 6 | 7 | 1 | 3 | 5 | 4 |
| Existing + | MD e+p | 7.07 | 6.54 | 6.13 | 6.93 | 7.33 | 6.37 | 6.40 |
| Planning | ranking | 2 | 4 | 7 | 3 | 1 | 6 | 5 |

[1] The symbols ↑ and ↓ indicate the rise and fall, respectively, of MD compared with the existing stage.

1. Determination of overall matching degree

In the existing stage, the rankings for MD showed that Baoying had the best matching degree of greenspace form with water system, followed by Xinghua and Gaoyou; Jianhu and Dongtai were lower, while Haian and Jiangyan had the worst matching degree. In the future plans, the seven cities, ranked from highest to lowest, were: Jianhu, Xinghua, Baoying, Haian, Jiangyan, Gaoyou and Dongtai. The matching degree for Jianhu was the best, while that of Dongtai was the worst.

If we consider the rankings for both the existing and future plans, Baoying and Xinghua had the best matching degree of greenspaces and water systems. These two cities had superior existing conditions. Xinghua's greenspace was substantially improved in the future plans. Although Baoying's two future plans indicators were reduced, which made it drop two places in the rankings, it still occupied the best position in the evaluation. The matching degree of Dongtai was the worst because of its average conditions at the existing stage. It was the only city where the scores on all secondary indicators in the future plans decreased.

2. Impact of greenspace plans on matching degree

Based on the analysis of the overall changes between the existing and future plans scores, we obtained the following conclusions. From the perspective of matching greenspaces with water systems, Baoying and Xinghua scored the highest, while Dongtai and Jiangyan scored the lowest. The MD rankings of the four cities (Jianhu, Haian, Xinghua and Jiangyan) increased due to improved greenspace indicators in greenspace development and planning. Jianhu and Haian showed the fastest rise in matching degree rankings, which was mainly due to the improved plans and arrangement of WGS and NGS. On the contrary, the other three cities (Dongtai, Gaoyou and Baoying) retreated in

their rankings. The decline in Dongtai and Gaoyou was particularly obvious. The sharp declines in NGS and WGS for Dongtai and Gaoyou were the main reason for the significant decrease in their rankings. This shows that the characteristics of urban water networks were not appropriately considered when planning the greenspaces of these two cities. In addition, development and construction of waterfront and near-water greenspaces were not taken seriously. In short, rational planning of greenspace can have a great impact on the matching degree.

### 4.1.2. Correlation and Change Regularity between Matching Degree and Various Indicators

The correlation analysis results showed a significant positive relationship between matching degree and primary indicators, in which the relationship between WGS and MD is the highest (r = 0.877, $p < 0.01$), followed by UGS (r = 0.739, $p < 0.01$) and NGS (r = 0.590, $p < 0.05$). In all cities, the change in matching degree between the existing stage and planning stage was consistent with the change in NGS, while there are a few exceptions for UGS and WGS. The scores rankings of WGS for Baoying and UGS for Gaoyou increased, but their total MD rankings decreased, especially for Gaoyou, which decreased the most.

The scores of UGS and WGS presented a remarkable positive relationship (r = 0.812, $p < 0.01$). No matter whether in the existing stage or planning stage, the changes in UGS and WGS indicators in all seven cities were consistent. However, there were some differences in the existing NGS. The rankings for NGS in Xinghua, Gaoyou, Dongtai and Jianhu were opposite to UGS and WGS. Jianhu had the lowest score and Dongtai had the highest, indicating that, although Dongtai had a low matching degree of UGS and WGS, the existing construction of NGS was relatively good.

The NGS in the planning stage was basically consistent, which showed that near-water greenspace was relatively consistent in future greenspace planning to fulfill the new "Park City" concept in Chinese urban development [39].

### 4.1.3. Some Correlation between Secondary Indicators

Regarding the secondary indicators for UGS, the rankings for greenspace area ratio (GAR) and mean patch area (AREA_MN) in the existing stage were basically the same as in the planning stage, indicating that there was a positive correlation between GAR and AREA_MN (r = 0.648, $p < 0.05$), and a negative correlation was found between GAR and patch density (PD) for UGS (r = −0.653, $p < 0.05$). In terms of the secondary indicators for NGS, the changes in area ratio (AR) and length ratio (LR) in both the existing and planning stages were completely consistent in all cities and showed a remarkable positive correlation (r = 0.945, $p < 0.01$). That is, if the area ratio increased or decreased, the length ratio also increased or decreased accordingly. The secondary indicators for WGS, PD and AREA_MN showed significant negative correlations (r = −0.956, $p < 0.01$). The common denominator was that these two indicators may be negatively correlated when the total amount of greenspace remains unchanged, but there will be other possibilities when the total amount of greenspace changes. Meanwhile, there are significant positive relationships for the same secondary indicators between different types of greenspace; specifically, the PD, AREA_MN and ENN_NN for UGS were all positively related with those for WGS (r = 0.947, 0.947 and 0.814, respectively, $p < 0.01$). Remarkable correlations were also found in different secondary indicators for different types of greenspace; for example, the GAR for UGS had a negative correlation with PD for WGS (r = −0.662, $p < 0.01$) and positive correlation with AREA_MN for WGS (r = −0.662, $p < 0.01$). It indicated that the spatial forms of UGS and WGS are similar to some extent.

### 4.2. Deficiencies and Unexplainable Results

Several relationships in the results are difficult to explain. Some indicators were completely consistent with changes in the seven cities, but the correlations or changes in other indicators were only found in some cities. This leads to a situation that cannot be well-

explained. Among the three groups of secondary indicators, in addition to the relatively consistent variability in NGS, the UGS and WGS indicators revealed the complexity of change (see Table 6). For example, the negative correlation between green patch density and green area appeared when the total amount of greenspace remained unchanged. However, it is not clear whether this relationship remain unchanged due to the increase or decrease in the total amount of greenspace.

In fact, there were inconsistencies between UGS and WGS in this study. Although the scores of PD and AREA_MN indicators for all cities showed a remarkable negative correlation (r = −0.978, $p < 0.01$), there were a few cases that the changes in PD and AREA_MN were the same. This is surprising given that the nature of these two groups of indicators was the same. As the total amount of UGS or WGS changed in the seven cities, the relationship between changes was also complex, and there are various possibilities. We tried to conduct an analysis on the possible relationship between them and found that most cities showed a negative correlation. Only the changes in PD and AREA_MN in Dongtai and Jiangyan were the same, but it could be explained (see Table 11). Furthermore, we found that this situation occurred when the difference between the existing and future plans scores was tiny (D_value < 0.05). We infer that there may be a threshold causing this situation, which is worthy of further study. Fortunately, neither of the unexplained cases happened. The changes in the relationship between indicators caused by the increase or decrease in the total amount of greenspace reflected the changes brought about by various greenspace plans.

**Table 11.** Abnormal situations and possible reasons.

| The Greenspace Area Increased↑ | | | | | |
|---|---|---|---|---|---|
| PD | AREA_MN | Rationality | Change Characteristics of Greenspace | UGS | WGS |
| ↑ [1] | ↑ | Explainable | Increased amount, especially the greenspace, which became larger than A [2]. | Dongtai, Jiangyan | |
| | ↓ | Reasonable | Increased amount, especially the greenspace, which became smaller than A. | | Baoying |
| ↓ | ↑ | Reasonable | Decreased amount, but increased greenspace, which became larger than A, indicating that more greenspace was merged together. | Xinghua, Gaoyou, Haian | Xinghua, Haian |
| | ↓ | Unexplainable | / | | |
| The greenspace area decreased↓ | | | | | |
| PD | AREA_MN | Rationality | Change characteristics of greenspace | UGS | WGS |
| ↑ | ↑ | Unexplainable | / | | |
| | ↓ | Reasonable | Increased amount and decreased area, indicating that more greenspace smaller than A was planned. | Jianhu, Baoying | Jianhu, Jiangyan |
| ↓ | ↑ | Reasonable | Decreased amount, but increased greenspace, which became larger than A | | Gaoyou, Dongtai |
| | ↓ | Explainable | Decreased amount and area, especially the greenspace, which became larger than A | | |

Note: [1] The symbols ↑ and ↓ indicate the rise and fall, respectively, of the corresponding indicators, compared with the existing stage. [2] A is the average area of the greenspace at the existing stage.

Some cities performed abnormally in the evaluation. For example, in the UGS for Dongtai, all three indicators increased, but only ENN_MN decreased. Moreover, the decrease in range exceeded the sum of the increases in the other three indicators. That is, the scattered distribution of UGS patches in the future plans eventually led to the decline in the overall score for UGS, which is quite unreasonable from the standpoint of planning practices. Another point is that the existing NGS in Dongtai had the highest score and had obvious advantages compared to other cities, but it ranked last in the score for future plans.

We can infer the reason by comparing the scores of various specific indicators. Compared with the existing NGS in Dongtai, the NGS in other cities had greater room for improvement and had been greatly improved in the planning stage, relatively speaking. On the contrary, the score for NGS in Dongtai decreased a great deal. Generally speaking, on the one hand, the distribution of green patches was relatively scattered. On the other hand, due to the good existing conditions, it was difficult to optimize and adjust the large

space for NGS in the planning stage, resulting in the final overall score for Dongtai City. This is still an issue to be discussed in depth.

In addition, for WGS, we can see that there were substantial differences between the future plans and existing data. This may be because the scoring method of this study compared seven cities. The scores for each city reflect its relative ranking among the seven cities. Therefore, the large differences between the future plans and existing scores does not necessarily refer to the actual differences between the plans and existing stages but instead reflect the large changes in the relative rankings between the two. There may have been some improvements in actual indicators, but to a lesser extent than in other cities.

### 4.3. Issues to Be Further Studied

Selection of indicators of greenspace is a core problem. Due to the different research objectives and scales, the indicators were not uniform [40]. Yunfei Jiang et al. investigated the impact of spatial structure between blue- and greenspaces on mitigating UHI effect using indicators including location of greenspaces, connectivity degree of the blue–green ecological network, river width and the distance of the waterfront green space from the riverbank [30]. It was revealed that a large green patch with high green coverage and connectivity degree and that was distributed in the leeward direction of the river corresponded to the lowest LST [30]. Although they are reasonable based on theory, there are intractable issues in practice considering the complicated situation of urban construction. Future research should determine which indicators could be simplified and combined to remove redundancy and identify the appropriate weight of the various indicators. This should be consistent with theory and feasible practices. To facilitate planning practices, we also need more effective, concise and operable evaluation tools. Therefore, evaluation methods should be studied further, including optimization of the evaluation indicators, convenient and accurate acquisition and mapping of existing data and analysis and evaluation of various greenspace functions.

It is essential to integrate ecology with urban planning and design [41,42]. Research on the correlation, or matching degree, of greenspace forms with water systems needs to focus on the linear biological ecotone space between waterbody and greenspace [32,43]. Future research should also focus on greenspace development and planning in rich water networks. A riverbank with a natural transition from waterbody to greenspace is ideal, so it is an important evaluation indicator of the naturalness of the riverbank [44,45]. However, within urban built-up areas in the study area, most of the current riverside zones in these cities were usually in the form of a vertical artificial retaining wall or artificial block stone-slope revetment. It is a pity that the naturalness of the riverbanks cannot be compared because such detailed data could not be provided in the urban greenspace system planning at the city level despite the fact that existing data for the waterfront were available.

Furthermore, we need to know what kind of biodiversity is most suitable in future frameworks. The typical habitat of WGS needs to be studied further, including aquatic plants, riverbank vegetation, birds and water quality [46–49]. Although some data can only be obtained from a large number of field observations or long-term ecological positioning research, these in-depth studies can expand our understanding of the biological ecology of the relationship between urban greenspaces and water systems rather than just planning for recreation, sports and landscape improvement.

## 5. Conclusions

There are abundant urban rivers in water network areas. The small cities in the study area can almost be called "water towns", which are still in development. The rivers and waterfront greenspaces in some urban built-up areas still retain a certain natural status. The main way of urban planning and construction in China is to retain the rivers and build riverside greenspaces. To achieve that, urban greenspace system planning and urban landscape planning require a performance evaluation tool to effectively resolve and improve problems and deficiencies in planning practices.

In order to address this problem, we developed a simple evaluation system to evaluate the matching degree between greenspaces and water systems at both the existing and planning stages. The framework can help to evaluate the matching degree of the urban greenspace to the water system to reflect the characteristics of the planning, layout and construction of urban greenspace with a dense water network. Further, analyzing the change in matching degree at the existing and planning stage can filter the primary influence factors, which are instructive to morphologically optimize urban greenspace planning. Particularly, rational urban waterfront greenspace construction could improve the ecological habitat quality of this unique ecotone in a city with a dense water system. This work is conducive not only to an in-depth understanding of the pattern of mutual integration of urban greenspace with urban water system but also to the planning practices of urban greenspace system planning and urban design.

**Author Contributions:** Conceptualization, X.-J.W.; Data curation, X.W. and X.C.; Methodology, X.-J.W. and X.W.; Software, X.W.; Supervision, X.-J.W.; Visualization, X.W.; Writing—original draft, X.-J.W., X.W. and X.C.; Writing—review and editing, X.-J.W., X.W., X.-J.W. and X.W. are co-first authors because of their equal contributions to the article. All authors have read and agreed to the published version of the manuscript.

**Funding:** This research was funded by the National Natural Science Foundation of China (50978054 and 51878144).

**Institutional Review Board Statement:** Not applicable.

**Informed Consent Statement:** Not applicable.

**Data Availability Statement:** Some or all data, models or codes that support the findings of this study are available from the corresponding author upon reasonable request.

**Acknowledgments:** We wish to thank the editor and anonymous reviewers for excellent comments and suggestions. We also thank Hao Zou, Mengjun Hu and Mengying Shen of Southeast University for data preprocessing of the planning maps, and Yu Gan for help regarding visualization.

**Conflicts of Interest:** The authors declare no conflict of interest.

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
