# Peer review of "Morphological Suitability Analysis of Urban Greenspaces with Rivers: A Case Study of the Lixiahe Riverine Area"

_sustainability, doi:10.3390/su142013266_

Round 1
Reviewer 1 Report
The article presents an interesting comparison of the park systems of several Chinese cities. Authors study their association with the rivers that run through these cities and compare the current state and future planning. Although it has more than enough scientific quality, I think the authors should follow the following guidelines before publication:
L 107: Please include here the meaning of the acronyms that are introduced in the methodological scheme. Right now, they are explained below (L 170-183).
L 127: (first column, third line): Please clarify that UGS stands for Urban GreenSpace. The first reference to this acronym is below (L 156).
L 132-133: "The existing data [...] by the Baidu Map." Please include reference to these sources in bibliography, link or endnote.
165-166: "NGS is [...] water system." Please revise the sentence because it doesn't seem to make sense.
L 171-172: "This is the most [...] evaluation in China." Please include reference to justify this statement.
L 185: Please separate further the three squares in the image, and place below each of them the type of greenspace being represented (UGS, WGS or NGS).
L 205: As a tip, I think you should include the maps of the cities that are being discussed with the GS classified, as they would help to better understand all the subsequent explanation.
L 345-348: Delete paragraph.
L 540-544: Please check that this format is according to editorial lines.
I hope that these changes will help to improve the paper and that it will be very successful. My congratulations to the authors.
Author Response
Responses to Reviewer 1
Thanks so much for taking your time to review this manuscript. We really appreciate all your generous comments and suggestions. Please find revisions in the re-submit manuscript. And here are responses to your comments:
- L122-123 The meaning of the acronyms have been added in the note of the Figure 1.
- L148 The UGS has been changed into urban greenspace in Table 1.
- L154&174 The link of Baidu Map has been added.
- L196 The sentence about the definition of NGS has been modified and moved into Table 4 (second column, fourth line).
- L190 The reference has been include to justify the statement “ This is the most [...] evaluation in China”.
- L263 The illustrations of the three types of greenspace have been modified in the Table 4.
- L509 Thanks for the tips. However, all kinds of greenspaces maps of seven cities are too much for the article. And this section are mainly focus on the statistical comparison from the global perspective, which may not so intuitive in the whole maps. To make up for that, we have selected a few illustrative examples to clarify the problem in Table 9 (L509). Hope it’s helpful.
- L468 The paragraph has been deleted.
- L715 The format of author contributions has been changed according to the editorial lines.
Reviewer 2 Report
Sustainability- 1865374 Morphological Suitability Analysis of the Urban Greenspace with Rivers: A Case study of the Lixiahe riverine area
Introduction
Where is the literature review of similar studies?
Materials and Methods
Line 176- in complete sentence- “in the meantime”?
Results
Lines 206- 223: It would be good to initially spell out the variables ENLL-MN, GAR, PD, AREA_MN when they are first mentioned in this section – otherwise the reader will have to search for the meaning. It also would help to do the same at the bottom of tables 5 & 6 for the same reasons.
Lines 321 & 322 & Line 334 in Table 7; Is Unit_Leng really Unit_lenght?
Discussion
Authors need to compare findings with any other similar studies
Conclusions
Lines 536-538; Authors need to discuss how this kind of research is “conducive to the accumulations of theories and methods for urban greenspace planning”. This needs much more discussion.
Author Response
Responses to Reviewer 2
Thanks so much for taking your time to review this manuscript. We really appreciate all your generous comments and suggestions. Please find revisions in the re-submit manuscript. And here are responses to your comments:
Introduction
Lines 89-95 The literature review of similar studies has been added in the last paragraph of the introduction.
Materials and Methods
Line 268 The sentence has been modified.
Results
Lines 303 The full name of the variables has been added when they are first mentioned in the section as well as in all Tables and figures.
Lines 445 & 446 & Line 422 in Table 8 Yes, Unit_Leng is the abbreviation of Unit_length.
Discussion
Lines 646-654 Because the study of the similar topic that focus on the planning and practice is rare, there are not so much comparison with other findings. To make up for that, we have compared the literature with similar selection of indicators, hope it’s helpful.
Conclusions
Lines 705-712 More discussion about the “This work is not only conducive to an in-depth understanding of the characteristics of urban greenspace” has been added.